# Pharmacist Administration of Long-Acting Injectable Antipsychotics to Community-Dwelling Patients: A Scoping Review

**DOI:** 10.3390/pharmacy11020045

**Published:** 2023-02-27

**Authors:** Andrea L. Murphy, Sowon Suh, Louise Gillis, Jason Morrison, David M. Gardner

**Affiliations:** 1College of Pharmacy, Dalhousie University, Halifax, NS B3H 4R2, Canada; 2Department of Psychiatry, Dalhousie University, Halifax, NS B3H 2E2, Canada; 3W.K. Kellogg Health Sciences Library, Dalhousie University, Halifax, NS B3H 4R2, Canada

**Keywords:** long-acting antipsychotic, pharmacist, review, injections, scope of practice, health services accessibility

## Abstract

Long-acting injectable antipsychotics (LAIAs) have demonstrated positive outcomes for people with serious mental illnesses. They are underused, and access to LAIAs can be challenging. Pharmacies could serve as suitable environments for LAIA injection by pharmacists. To map and characterize the literature regarding the administration of LAIAs by pharmacists, a scoping review was conducted. Electronic-database searches (e.g., PsycINFO, Ovid Medline, Scopus, and Embase) and others including ProQuest Dissertations & Theses Global and Google, were conducted. Citation lists and cited-reference searches were completed. Zotero was used as the reference-management database. Covidence was used for overall review management. Two authors independently screened articles and performed full-text abstractions. From all sources, 292 studies were imported, and 124 duplicates were removed. After screening, 13 studies were included for abstraction. Most articles were published in the US since 2010. Seven studies used database and survey methods, with adherence and patient satisfaction as the main patient-outcomes assessed. Reporting of pharmacists’ and patients’ perspectives surrounding LAIA administration was minimal and largely anecdotal. Financial analyses for services were also limited. The published literature surrounding pharmacist administration of LAIAs is limited, providing little-to-no guidance for the development and implementation of this service by others.

## 1. Introduction

Long-acting injectable antipsychotics (LAIAs) are prescribed in mental health care for patients experiencing persistent and serious mental illnesses (SMIs) (e.g., schizophrenia, schizoaffective disorder [1]). They are often prescribed in the context of patient nonadherence to oral antipsychotics (OAPs) [2,3,4], but have broader benefits for health outcomes [5]. Treatment adherence for SMIs is critical, given this population is at risk of poorer health outcomes including premature death [6]. Discontinuation of OAPs is common [7,8,9,10], thus making the LAIA formulation an important adherence-promoting option. In addition to adherence, LAIAs have positive benefits on outcomes such as mortality, hospitalization reduction, quality of life and functioning, and relapse prevention, compared to OAPs in patients with schizophrenia [2,7,11,12,13,14]. Benefits have also been shown in specific subgroups. In early psychosis, LAIAs have shown promise for adherence outcomes, relapse reduction, and symptom improvements [15]. Long-acting injectable antipsychotics may also be beneficial for outcomes in those experiencing schizophrenia and co-morbid illnesses (e.g., substance-use disorders) [16]. In patients with bipolar disorder, LAIAs have been studied in mania prevention with good outcomes (e.g., preventing mania recurrence) [17].

Additional justification for LAIAs beyond adherence and other health outcomes derives from potential economic benefits. Lower or neutral impacts on overall costs are possible to achieve when considering trade-offs with preventing poorer outcomes (e.g., hospitalizations, mortality) against higher medication costs and fees for injection related to the LAIA formulation [12,18].

The prevalence of LAIA use is low in Canada. Agid et al. [19] report Canada’s rate of LAIA use as 6.5%, from prescription claims in those aged 16 to 35 years of age with a diagnosis falling within a spectrum of schizophrenia-related disorders [19]. This is in contrast to international counterparts, where the prevalence of LAIA use ranges from 15 to 80% [19,20]. Low uptake of LAIAs is multifactorial with several barriers to their use from patient and prescribers’ perspectives [21,22,23,24,25]. Notably, there may be ambivalence and knowledge gaps with prescribers in which they doubt, or are unaware of, whether LAIAs can be used in certain populations (e.g., first-episode psychosis) and make assumptions that patients prefer oral medications [24]. This is of particular relevance, as patients have been shown to have positive attitudes towards LAIAs, recognize the benefits (e.g., reduced hospitalizations) [26], and may be more accepting of LAIAs as part of their treatment regimen if recommended by their clinicians [25]. Recent findings from a discourse analysis demonstrate that rejection of LAIAs by patients may be influenced by the ambivalence of psychiatrists [22].

Access to LAIAs can be a significant barrier. Travelling to receive the LAIA injection can be burdensome for patients [26] and transportation difficulties can impact persistence with treatment [27]. Living in rural or remote areas where mental-health-care services can be sparse or practically inaccessible, due to long waitlists, can limit access [28]. Psychiatrists or specialty services that start LAIA treatment may encounter difficulty with transfers of care, as primary-care providers may lack the competency or capacity to provide this service [29].

Community pharmacies may offer a solution to many of the barriers to accessing LAIAs. In Canada, pharmacies are geographically well positioned with good density to meet the needs of many rural and remote communities [30,31,32]. A recent analysis also shows that pharmacies in Canada have longer operating hours in communities where there is a lower median income and higher service need [32]. Providing LAIA injection services with pharmacists may facilitate access for patients closer to their homes. Individuals with SMI are already engaging with community pharmacists in many parts of the world, given pharmacists’ roles in medication-supply management and rates of oral medication use and polypharmacy in these groups [33,34].

Community pharmacists are an accessible group of health-care professionals with a growing scope of practice. There is increasing awareness, development, and implementation of enhanced roles for community pharmacists in mental health care [34,35,36,37,38,39,40,41,42,43]. The research in this area includes a breadth of services, many that focus on the strengths of pharmacists’ competencies in pharmacotherapeutic management, and others that are complementary to a patient’s comprehensive care. For example, screening for mental illnesses (e.g., depression) [44,45,46] and substance-use disorders (e.g., alcohol) [47,48] has not traditionally been thought of as part of a pharmacist’s role, but can help to identify drug-therapy problems. These services augment and facilitate routine medication management, and, when implemented, they are acceptable to patients. The evidence from patient outcome data is also supportive of pharmacists’ roles in various mental-health and substance-use-disorder therapeutic areas. For example, pharmacist-managed depression produces similar outcomes to treatment as usual in primary care [49]. Other areas are identified on an ongoing basis where there is a need to have pharmacists more engaged with service delivery (e.g., naloxone) [50]. Moreover, members of the public view the community pharmacist positively in these roles, and see the environment as an appropriate venue for mental-health promotion [51].

In Canada, supportive legislative frameworks in many provinces have allowed for an expanding scope of practice of pharmacists to include the injection of medications [52]. This can include LAIAs, but to date, there is limited information about the existence of such services in the Canadian context, and more information is available from the United States (US). The first step in preparation for the development and implementation of a community-pharmacy-based LAIA service to improve access in Canada would be to map and characterize the literature regarding pharmacist and patient roles and their experiences with such services.

A scoping review was conducted to map and characterize how pharmacists’ and patients’ roles and experiences with respect to LAIA administration by pharmacists have been studied. Scoping-review methods were most appropriate for the review, given the desire to map the nature of evidence available, as opposed to a systematic review or meta-analytic approach in which outcomes would most likely be quantitatively analyzed. For example, if an effect size for one or more interventions was the desired outcome, a meta-analysis of data would be more appropriate. Scoping reviews allow for mapping of the evidence, regardless of the quality of the studies, and allow for clarification of the research gaps and concepts [53,54].

## 2. Materials and Methods

The scoping review was conducted using six stages (Table 1) modeled after Arksey and O’Malley [55] and following best practices for scoping reviews as guidance, including using the Preferred Reporting Items for Systematic Review and Meta-Analyses extension for Scoping Reviews (PRISMA-ScR) [56] and recommendations from the Joanna Briggs Institute (JBI) [53,54,56,57,58,59,60,61,62]. A protocol was not previously published or registered related to this review.

### 2.1. Inclusion and Exclusion Criteria

Studies or publications that examined pharmacists’ and patients’ experiences and roles with respect to pharmacist administration of LAIAs were eligible for inclusion. All publication types were eligible (e.g., abstracts, theses, full papers). Experiences and roles were broadly defined, so as to be all-encompassing. Pharmacists were required to administer the LAIA and there were no exclusions regarding the indication or associated medical condition related to the injection. Those reporting pharmacist-administered LAIAs in the context of hospitalized patients were excluded. Studies were not excluded based on language or dates.

### 2.2. Concepts

The concepts included the formulation (e.g., long-act, depot, LAI, long-acting drug), route of administration/ dosage form (e.g., inject, intramuscular injection), profession (e.g., pharmacist), and drug type (e.g., antipsychotic, neuroleptic, paliperidone, aripiprazole, etc.).

### 2.3. Context

The setting for studies was not limited to any geographical location, and included community-based settings that were urban, rural, and remote settings. Studies reporting on outpatient clinic were included if affiliated to a health centre or hospital, provided that the patients receiving the services were not actively staying in hospital. Hospital-based administration for patients who were categorized as inpatients (i.e., staying overnight in hospital for periods of time) were ineligible. International literature was acceptable.

### 2.4. Search Strategy

Our search of electronic databases included PsycINFO, Ovid Medline (see Appendix A), Scopus, and Embase with searches completed by 15 July 2022. Searches were not limited by dates or by language. The primary search strategy was developed in Medline by a health-sciences librarian (LG) in collaboration with other team members (ALM, DMG, JM, SS), and peer reviewed by another librarian using the Peer Review of Electronic Search Strategies (PRESS) approach [63,64].

To start the development of the search strategy, a concept map was used to track idea generation with search terms on each concept of the research question [65]. Additional search terms were identified through reviewing the indexing and title/abstract keywords of known relevant articles. The search strategy was adapted for each included information source and tested to determine its recall of known relevant studies.

To supplement our primary search, we also conducted a search of ProQuest Dissertations & Theses Global and Google, using a strategy adapted from our primary Medline search. We set a preliminary stopping criterion of 200 records to view in Google, with flexibility in viewing more results based on what was found in the first 200 records. We also screened the citation lists of key articles, and conducted cited-reference searches with Scopus.

### 2.5. Reference-and-Data Management

Search strategies were documented, and Zotero [66] was used as the reference-management database. References retrieved from our searches were uploaded to Covidence for overall scoping-review management [67]. The final complement of studies was determined following title and abstract screenings, followed by full-text screenings. Microsoft Excel was used for data extraction. It was also used to examine patterns among the data (e.g., year of publication, countries of publication).

Two authors independently conducted screenings for the title and abstract (ALM, SS) and full-text stages (ALM, SS). Full-text articles screened for inclusion were subsequently abstracted by two reviewers (ALM, DMG). Data were summarized according to the data extracted numerically (e.g., number of publications) and information was also categorized with respect to potential facilitators and barriers to pharmacist-administered LAIAs.

## 3. Results

### 3.1. Study Inclusion

From all sources, 292 items were imported for screening (Figure 1). The search of electronic databases produced 130 from Embase, with fewer from the others (Medline (*n* = 31), Scopus (*n* = 83), and PsycINFO (*n* = 9)). Duplicates (*n* = 124) were removed in Covidence. There were 168 studies screened at the title-and-abstract stage, with 147 deemed irrelevant. Twenty-one studies were assessed in the full-text stage. Eight were excluded, with five not including specific information about pharmacists or patient roles and/or experiences, two used other disciplines and not pharmacists as injectors, and one contained no new information and cited papers already collected. After the screenings were completed, 13 studies [68,69,70,71,72,73,74,75,76,77,78,79,80] were included for data extraction.

### 3.2. Data Extraction

Data were extracted using the following fields: authors, year of publication, country of publication, type of publication, patient demographics (sex, age), patient and/or pharmacist outcomes, settings (e.g., clinic, pharmacy), and facilitators and barriers related to LAIA administration (e.g., time, competencies, finances). The two reviewers met to review and discuss the findings from the data-extraction process, and no major discrepancies regarding content or concepts were uncovered.

### 3.3. Characteristics of Included Studies

#### 3.3.1. Country and Year of Study

All publications were located in the US, and all but one article [70] had been published since 2010, with three being the most published in any one year (i.e., 2012) [75,76,78].

#### 3.3.2. Types of Publications

All publications were characterized as retrospective in nature, and no studies with a prospective design were identified. Two citations were available in abstract form [77,80]. One citation was the report from a Post-Graduate Year-1 Pharmacy-Residency-Program project [71]. Many publications (n = 7) [69,70,71,75,76,77,79] were categorized as commentaries or narrative descriptions of “practice innovations” or “practice experiences”, with pharmacists performing LAIA services. Seven publications (Table 2) reported data on at least one outcome.

#### 3.3.3. Antipsychotics Included

Paliperidone was the most-reported LAIA across the included publications, followed by aripiprazole (Table 3).

#### 3.3.4. Patient Characteristics

The proportion of males was higher than females when the sex of patients was presented in studies [68,70,72,74,78]. Mascari et al. [73] was the exception, with more females completing their survey. However, there were only eleven of forty receiving the LAIA service who responded to the survey, and of these, only nine provided demographic information. A broad range of ages was represented among studies that provided age information with the majority being over 18 years of age, with one case exception that was described as a “fairly large 17-year-old” [79].

### 3.4. Outcomes

#### 3.4.1. Adherence

Two articles [68,72] reported analyses from databases with the community pharmacy as the setting for injection. Pharmacist administration of LAIAs had positive impacts on adherence as measured by the proportion of days covered (PDC), which is the number of unique total days with medication coverage (numerator) divided by the days in the measurement period (denominator) [68,72]. These studies reported results from complex, organized programs from companies (i.e., Janssen and Albertsons Companies).

#### 3.4.2. Attitudes and Satisfaction

Three studies [73,74,80] measured patient or pharmacist satisfaction using surveys (Table 2). The patient voice from the retrieved evidence was limited to 115 patients [73,74].

Patients were satisfied with pharmacist administration of LAIAs and positively viewed other attributes of the pharmacists (e.g., trustworthiness) and the setting (e.g., privacy) [73,74]. Convenience was viewed positively [74] and this was also identified in a specific study conducted during the COVID-19 pandemic [73], although the sample was small, with eleven respondents. Survey response rates from patients were generally low (i.e. 27.5% [73], 14.5% [74]). The patient perspective reported by Mooney et al. [74] was in the context of receiving services within a “speciality care” program of the Albertsons Companies.

Gupta [80] reported on attitudes and perspectives of pharmacists with LAIA-injection services. The report was limited in length in its abstract form. In the sample of 200 community pharmacists, 9% (*n* = 18) had previous LAIA-administration experience. Views were mixed towards LAIA administration, regardless of previous LAIA-injection experience, with 44% (*n* = 88) reporting a positive attitude towards administration and 42% (*n* = 84) reporting a negative attitude.

#### 3.4.3. Financial Impact

Phan and VandenBerg [76] provided the most detailed financial information of pharmacists’ roles in an LAIA clinic. The retrospective review included information from seven patients seen over three months. The clinic was established to shift LAIA administration from hospitalization to the outpatient clinic. Financial information related to LAIA services was provided in other publications, but it was often limited by sample sizes and discussions of dispensing fees [71,79] or embedded within a larger model of service delivery, as was the case with Tallian et al. [78]. Separating costs related to LAIA interventions by pharmacists was challenging when this was combined with other services.

### 3.5. Facilitators and Barriers relating to LAIA Administration

#### 3.5.1. Safety

The most prominent barriers, identified by 61% (*n* = 122) of the sample in Gupta [80], were safety concerns. Three different groups were listed regarding safety concerns: customers, self, or patients. There was no further elaboration of positive and negative attitudes, aspects of safety that were considered, or the specific safety concerns.

#### 3.5.2. Time and Scheduling of Injections

Having a designated time for the injection was the most important method of decreasing administration barriers, as identified by 76% of respondents in the survey by Gupta [80]. Tallian et al. [78] provided details on the role of the pharmacist in a pharmacist–psychiatrist collaborative medication-therapy management (MTM) clinic, and commented that for the development of depot administration clinics, more pharmacists and/or time would be required.

The use of appointments or scheduled injections was discussed in 10 papers [68,69,70,71,72,73,74,76,77,78]. One pharmacy’s appointment system described in 2010 [77] was changed to an on-demand system, since patients found it difficult to keep appointments [69]. Phan et al. [76] also reported the need for flexibility with scheduling, and good discharge planning when patients transition from a hospitalization to community-based care. Several publications provided information on appointment times when injections were scheduled, but these appointments also included other assessments. Giles [70] indicated that appointments were 10 to 30 min, with a 17 min average. Similarly, Mascari et al. [73] reported 20 min and Mooney et al. [74] indicated approximately 30 min appointments. Tallian et al. [78] reported that new patients had 60 min appointments and 30 min for return visits, but that pharmacists spent 26 min on average with patients in appointments. However, within this study reporting, there was no clarity with respect to what proportion of patients received LAIAs and their appointment length for this service.

#### 3.5.3. Settings

Three studies [69,71,77] occurred in a community-pharmacy setting, and four [70,76,78,79] were classified as occurring in outpatient specialty clinics. Some descriptions of the setting made the distinction between community pharmacy and clinic difficult to decipher [69,77].

#### 3.5.4. Education and Training

Education and training programs for the pharmacists administering the injections were infrequently described. Three [69,72,74] of the 13 papers included training requirements of one specific group of pharmacies and reported that pharmacists were trained using both manufacturer’s and in-house-developed training programs. Gupta et al. [80] reported from their survey that 80% and 64% of the 200 pharmacist respondents identified live training with hands-on practice and instructional videos, respectively, as the most appropriate educational tools. “Continuing-education programs” were also identified as appropriate by 65% of the sample [80]. The specific opinions on education were not reported according to whether the pharmacist had experience with injection provision. In the study by Tallian et al. [78], the pharmacists were Doctor of Pharmacy prepared, required to have an acute-care or psychiatric-pharmacy residency, had board certification in psychiatric pharmacy, and needed to complete additional training such as a physical exam course.

#### 3.5.5. Collaboration

Collaboration with other health-care professionals was directly mentioned or implied in all articles except the survey of pharmacists’ attitudes towards LAIAs [80]. The most frequently mentioned collaborators were prescribers such as psychiatrists and other non-specialist physicians. The descriptions around procedures for collaboration were often limited, but some papers provided more details on the structure of the relationship between collaborators [78]. Some programs described their model as a team approach [70] or a team, along with significant independent work of the pharmacist [78]. These pharmacists had reliable access to disciplines (e.g., master’s-prepared social worker) that may not be consistently available to all community pharmacists for collaboration.

#### 3.5.6. Procedures for LAIA Administration and Ongoing Monitoring

The level of detail provided on the pharmacists’ procedures and activities during and around injections was variable among publications. A consent process for patients prior to the injection was specifically mentioned in four publications [71,73,74,78]. It was evident that pharmacists had access to patients’ bloodwork to support decision-making in three papers [70,71,78]. The monitoring of medication response and of adverse events was cited specifically in three papers [70,72,73], and would be an expected activity by most pharmacists as part of their scope of practice, but other activities were also discussed, such as pharmacists conducting the Clinical Global Impression Scale [71]. Tallian et al. [78] provided details on the role of the pharmacist administering LAIA injections, conducting physical assessments (e.g., vital signs), metabolic screening, and using scales to assess symptoms (e.g., Clinical Global Impression scale) and side effects (e.g., Abnormal Involuntary Movement scale). Mascari et al. [73] discussed pharmacists using a “standard questionnaire” for adverse reactions (e.g., hyperprolactinemia, extrapyramidal symptoms).

#### 3.5.7. LAIA Insurance and Injection-Fee Reimbursement

The studies reporting on the programs from Janssen and Albertsons Companies [68,72] included benefits investigations regarding medication coverage for patients. This service was not performed by the injecting pharmacist, and was conducted with the assistance of other, centralized personnel. Thompson [79] similarly reported on a clinic that hired a mediation-access coordinator to facilitate benefits and reimbursement, but injection fees were not specifically discussed. Bonner [69] reported that the absence of payment is common, and that options include seeking compensation from patients or using drug company/manufacturer programs. Kingsbury [71] provided third-party-payer information for the medication costs, but no injection-service fee was reimbursed. Mascari et al. [73] reported that the pharmacist medication-administration fees became reimbursed as part of a care model that included reimbursement of services from multiple providers. Similarly, Tallian et al. [78] reported that pharmacists billed for services based on time (i.e., minutes spent), similar to the process used by other health-care providers (e.g., psychiatrists).

A detailed breakdown of reimbursement was not provided by Phan and VandenBerg [76] in their financial analysis. However, they did indicate that waiting for prior authorization of medication coverage from insurers was reported as an important barrier in patient care. It served to deter from LAIA use in favor of continuation of oral medication.

#### 3.5.8. Legislative Framework

One publication by Oji et al. [75] did not directly report the roles of pharmacists, but focused on legislation. An email inquiry was sent in 2011 to US state boards of pharmacy requesting information on the legislative framework and protocols for LAIA administration by pharmacists. At the time of the inquiry, 21 states reported privileges of administration of injectable medications other than vaccines. Three specific examples of LAIA-injection practices were provided in brief, point form, with one of the examples already identified [77] through the scoping-review search. Bonner detailed pharmacist-legislated injections in 40 state, with 28 giving pharmacists broad authority to give injections without collaborative-practice agreements [69]. Thompson [79] indicated pharmacists began staffing the LAIA clinic when the Kentucky Board of Pharmacy added “administration of medications or biologics…” as part of the definition of the practice of pharmacy.

#### 3.5.9. Study Funding

The funding was not consistently reported in publications. Three teams reported their source of financial support for their work as grant funding [70,72,78]. Two articles cited pharmaceutical-industry funding or had authors from the pharmaceutical industry [68,80]. For the remaining articles, funding was not reported or could not be determined.

## 4. Discussion

The current research regarding pharmacist and patient experiences and roles with LAIA administration is limited, as demonstrated by the thirteen studies included, with no clear best practice or evidence-based approaches at this time. There are significant gaps in the literature regarding community-pharmacy operations/procedures for the service, education and training, reimbursement for services, collaboration models, and ongoing monitoring of care. Patient and health-system outcomes are very limited. There are also few studies on which to formulate the direction of future research regarding the best positioning of community-pharmacist-administered LAIAs in the patient journey (e.g., transitioning from inpatient to community care, community-based oral-to-injection switching, etc.). These limitations make it challenging for those seeking to adopt or modify best practices and other evidence-based approaches towards practice implementation. The corpus quality was also limited. No studies were found that included an active-comparator group to assess outcomes or to test hypotheses. Overall, the reporting and level of detail and description of interventions were inconsistent and mostly limited. A recommendation going forward is that the Template for Intervention Description and Replication (TIDieR) [81,82] be used by investigators when reporting the development and implementation of interventions for pharmacist-LAIA-service delivery. This would help to facilitate and explicate details regarding interventions that others may be able to replicate or adopt.

### 4.1. Complex Programs and Supports

The two papers measuring adherence and showing positive results were based on the programs from Janssen and Albertsons Companies [68,72]. These programs are complex, with infrastructure around the programs and tools (e.g., education and training), to support pharmacists in providing injection services. Differentiating the contribution of the pharmacist as injector to the outcome is challenging, given the extensive support provided by various personnel outside of the pharmacy context to patients in these programs in general (e.g., benefit investigation). Replication of such a service in a typical community pharmacy in many contexts, including a Canadian context, would likely be difficult without the support of additional staff and resources. Benefit and insurance exploration and management by community pharmacists can take a significant amount of time and present an opportunity cost, as these activities could take away from pharmacists’ clinical services with patients.

### 4.2. Pharmacists’ Perspectives on Services and Workload

Information about pharmacists’ perspectives on LAIA services was negligible. Importantly, no information was found regarding program implementation that included pharmacists’ evaluation of workflow and other procedures (e.g., on-demand versus appointment-based systems, etc.). One paper reported on the change in a system from appointment to on-demand [69], which may conflict with pharmacists’ perceived need to have injections occur at an allocated time [69] in order to be able to manage the range of on-demand services that occur daily in the community-pharmacy environment. Activities related to dispensing still consume a significant amount of pharmacists’ time, despite advances in scope of practice. Time for services such as injections (e.g., influenza) and general clinical services with patients has been shown to be less, because of dispensing-related tasks [83]. Appointment-based models (ABM) and related activities such as medication synchronization are increasing in popularity in the community-pharmacy setting [84,85,86]. The ABM has shown improvements in adherence for people taking medications for many long-standing health issues [86], although psychiatric diagnoses are less represented in this literature.

### 4.3. Education and Training

The educational background and training of the pharmacists engaged in LAIA administration was often not reported, as were their perspectives on education and training. Tallian et al. [78] was an exception in terms of the descriptive detail provided. The pharmacists working in the collaborative clinic with advanced training (i.e., board certification in psychiatric pharmacy, and a residency) would be considered exceptional, compared to the preparation of most community pharmacists.

### 4.4. Pharmacists’ Knowledge and Attitudes

Education and training programs would need to consider both elements of knowledge of and attitudes to SMIs and the specific skills required for injection of LAIAs. A recent systematic review by Crespo-Gonzalez et al. [87] reported on mental-health training programs for community pharmacists, pharmacy staff, and students, and included 33 studies. Training programs were varied across studies, but it was generally found that training programs made positive changes in important constructs related to stigma, attitudes, and knowledge. This may be particularly important with respect to the “safety concerns” and negative attitudes that were expressed in the small sample from Gupta et al. [80]. The data were too brief and limited to make extrapolations, and although the root of safety concerns and negative attitudes should not be conflated with stigma, it requires further explication, given previous research has identified stigma in pharmacy practice [35,88,89]. Reports of pharmacists’ satisfaction with LAIA services could also serve as evidence to counter this potential concern regarding stigma. The abstract by Singer [77] indicated that “patient and provider satisfaction surveys will be summarized” as part of the abstract, but the data were not presented in the abstract. Sixty-nine percent of pharmacists in Gupta et al. [80] reported satisfaction, which is less than some other studies regarding injection services for other products, mainly immunizations. For example, in a study by Hattingh et al. [90], 89.5% of their sample strongly agreed or agreed that one of the main reasons for providing pharmacist vaccination services was for the professional satisfaction of pharmacists.

### 4.5. Financial Considerations and Context

Financial benefits to the health-care system were reported [76], and can inherently occur with improvements in adherence; however, further economic considerations need to be explored. There is insufficient research from the existing publications to extrapolate whether investing in the development and implementation of a LAIA program would generate sufficient revenue to sustain such a program in the community-pharmacy context in countries such as Canada. All publications were conducted in the US, which is a barrier. Costs to the health-care system and medication-related insurance costs would differ significantly, compared to other countries.

### 4.6. Collaboration and Communication Procedures

Collaboration and communication are critical components of interdisciplinary mental-health care. Physicians were the most common collaborators reported, although other disciplines were mentioned. Various aspects of collaboration and communication would be useful and important to measure (e.g., frequency, modes, etc.) with the development and implementation of pharmacist LAIA-administration programs. Economic considerations around payment models for collaboration, consultations, and communication would also need to be explored and could be particularly important when disciplines are paid via different funding models (e.g., fee-for-service versus salaried).

## 5. Strengths and Limitations

This is the first scoping review to explore the topic of LAIA administration by pharmacists to community-based patients. There are significant gaps in the current literature, demonstrating that there is opportunity for more research to be conducted in this area. The information was very limited on this topic, not only in terms of the number of publications, but also in the quality of reporting, and thus, the quantity and quality of information that could be reported was also limited.

## 6. Conclusions

There is limited published information available to guide pharmacists in the development and implementation of pharmacist-led injecting of LAIAs. Significant gaps in the research exist. Higher-quality prospective research exploring outcomes related to pharmacists, patients, other health-care professionals, and health-system impacts is needed.

## Figures and Tables

**Figure 1 pharmacy-11-00045-f001:**
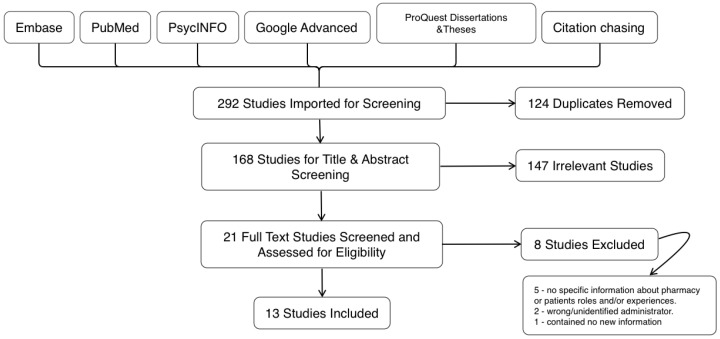
Preferred Reporting Items for Systematic review and Meta-Analyses extension for Scoping-Reviews [56] flow diagram for long-acting injectable antipsychotics administered by pharmacists for community-based patients.

**Table 1 pharmacy-11-00045-t001:** Scoping Review Stages [55] for Pharmacist Administration of Long-Acting Injectable Antipsychotics.

Stage	Purpose	Summary of Activities
1	Identify research question	Team meetings were held to discuss the review question, definitions, objectives, inclusion criteria, exclusion criteria, potential key terms, and search sources. The preliminary categories for the data-extraction tool were identified.
2	Identify relevant studies	The inclusion and exclusion criteria were finalized, as was the data-extraction template. The database searches were conducted. Searches were tracked and recorded and uploaded into COVIDENCE. Duplicates were removed.
3	Select studies	The inclusion and exclusion criteria were applied throughout the process of title and abstract screening and full-text screening.
4	Extract data	The data were extracted using the data-extraction template in Excel.
5	Collate, summarize, and report results	The data were summarized and prepared in written and table format.
6	Engage stakeholders	Knowledge translation will occur by engaging key stakeholders in both mental health care and pharmacy, to discuss the review findings.

**Table 2 pharmacy-11-00045-t002:** Database and survey studies of pharmacist roles in long-acting injectable antipsychotic (LAIA)-administration.

Author, Year	Study Design	Sample Analyzed for the Outcome	Outcomes
Benson et al., 2015 [68]	Retrospective database study	2659	Medication adherence
Lin et al., 2019 [72]	Retrospective database study	641	Medication adherence
Gupta et al., 2016 [80]	Survey	200	Pharmacists’ attitudes and barriers towards LAIAs
Mascari et al., 2022 [73]	Survey	104	Patient satisfaction
Mooney et al., 2018 [74]	Survey	11	Patient satisfaction
Phan and VandenBurg [76]	Retrospective, patient chart/database review/audit *	7	Financial impact
Tallian et al. 2012 † [78]	Retrospective, patient chart/database review/audit *	Unclear, fewer than 23 ‡	Number of patients co-managed, dropout rates, visit duration, billed minutes.

* The data source was assumed from how the data were presented, but it is unclear how the information was retrieved. † A “co-managed” service (i.e., collaborative with psychiatrists) was described, but stated that pharmacists administered LAIAs as part of the service. ‡ 23 patients were taking “antipsychotics”, but the number receiving injectable antipsychotics was not specified by the authors.

**Table 3 pharmacy-11-00045-t003:** Antipsychotics referred to from included studies *.

	Aripiprazole Lauroxil	Aripiprazole Monohydrate	Fluphenazine Decanoate	Haloperidol Decanoate	Olanzapine	Paliperidone Palmitate	Risperidone
Benson et al., 2015 [68]						X	
Giles 1976 [70]			X				
Kingsbury 2020 [71]				X		X	
Lin et al. [72]	X	X				X	X
Mascari et al. [73]		X		X		X	
Mooney et al. [74]		X				X	X
Oji et al. † [75]			X	X	X	X	X

* Six publications [69,76,77,78,79,80] referred to LAIAs as a group without identifying specific agents, and as such, are not reported under individual medications in the table. † Oji et al. [75] indicated medications in an Appendix regarding examples of expanded injectable programs and some LAIAs may have been indicated in more than one program.

## Data Availability

The summary of data as presented is representative of the dataset.

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
