# Peer review of "Pharmacist Administration of Long-Acting Injectable Antipsychotics to Community-Dwelling Patients: A Scoping Review"

_pharmacy, 2023, doi:10.3390/pharmacy11020045_

Round 1

Reviewer 1 Report

Thank you for the opportunity to read this systematic review. The topic is novel, and, as the authors say, interest is largely confined to the USA.

Where is the protocol for this review registered? A review cannot be accepted without prior registration.

Patient safety and adverse drug reactions are important aspects of any intervention and its management. I found no information on this. If there is none, then this is a serious limitation of the data available, and should be discussed.

The reference list is current and comprehensive.

The presentation of the review needs to be developed. Line 110 refers to 6 stages in a Table 1. Table 1 does not give this information.

The studies included in the review must be presented in full, with findings, in a table. An example is available in Sletvold et al 2022.

The strengths and limitations of each included study should be tabulated. One strategy for this is a ‘Risk of bias’ table.

How will the results of the search be made available to readers?

Methods

Please list the concepts used in the review.

Please clarify how and when search terms were combined.

It would be useful to contact authors where only abstracts were available. Papers may be forthcoming.

Results

The text is overly long, and without subheadings to guide the reader.

Some studies are described in detail, when they could be more succinctly tabulated. I would expect more synthesis and analysis.

Discussion

This is overly long, and without subheadings. The text repeats the results section, and should be curtailed.

Headings for ‘limitations of the data’ and ‘limitations of the review’ are needed.

Minor points

Authors undertaking tasks should be identified by initials.

Sletvold, H., & Jordan, S. & Olsen,R. M. (2022). Nurse-Led Interventions to Promote Medication Adherence in Community Care: A Systematic Review. In R. M. Olsen & H. Sletvold (Eds.), Medication Safety in Municipal Health and Care Services (Chap. 8, pp. 163–191). Cappelen Damm Akademisk. https:// doi.org/10.23865/noasp.172.ch8 open access

https://press.nordicopenaccess.no/index.php/noasp/catalog/book/172

Reviewer 2 Report

Thank you for the opportunity to review this paper. While it is an interesting study, I feel in its current form, it is not publishable. The methods section is lacking key details and the results as currently presented is all over the place and would benefit from some categorization/subheadings. Further comments are attached

Round 2

Reviewer 1 Report

Thank you for the quick response to our comments.

We need to see the responses to both reviewers in full.

As the authors say, there are differences between systematic reviews and scoping reviews with a systematic approach. However, all research should be registered. It is untrue to say that there is nowhere to register a scoping review.  

https://www.researchregistry.com/why-register

I do not think any unregistered review can be accepted. Before we reconsider, the authors must register, re-run the searches and resubmit.